# Glutathione S-Transferase May Contribute to the Detoxification of (S)-(−)-Palasonin in *Plutella xylostella* (L.) via Direct Metabolism

**DOI:** 10.3390/insects13110989

**Published:** 2022-10-28

**Authors:** Qiqi Fan, Jiyuan Liu, Yifan Li, Yalin Zhang

**Affiliations:** Key Laboratory of Plant Protection Resources and Pest Management, Ministry of Education, Entomological Museum, College of Plant Protection, Northwest A&F University, Yangling, Xianyang 712100, China

**Keywords:** *Plutella xylostella* (L.), (S)-(−)-palasonin, botanical insecticide, metabolism

## Abstract

**Simple Summary:**

*Plutella xylostella* is one of the most devastating pests worldwide due to resistance to a variety of chemical pesticides. Therefore, there is an urgent need for control alternatives. Our previous studies have found a plant-derived active substance (S)-(−)-palasonin (PLN) has a competent toxic effect on *P. xylostella*. However, we noticed cross-resistance between (S)-(−)-palasonin and other insecticides. We therefore hypothesized that metabolic resistance may be more important in (S)-(−)-palasonin resistance than target-site resistance. We investigated the contribution of detoxification enzymes in the metabolism of (S)-(−)-palasonin in *P. xylostella*. We found that the GST enzyme activity in the field strain of *P. xylostella* changed most significantly compared with the sensitive strain. We then evaluated the role of GSTs in detoxifying (S)-(−)-palasonin. The results show that the GST activities of *P. xylostella* increased significantly after exposure to (S)-(−)-palasonin. RT-qPCR shows that 19 of 20 GSTs genes were up-regulated after exposure to (S)-(−)-palasonin. Furthermore, the results of in vitro inhibition and metabolic experiments show that (S)-(−)-palasonin can competitively bind to GST, and that GSTd1, GSTd2, GSTs1 and GSTs2 have a capability to metabolize (S)-(−)-palasonin. This study contributes to the application of (S)-(−)-palasonin in the control of *P. xylostella* and to the resistance evaluation of botanical insecticides.

**Abstract:**

The control of *P. xylostella* primarily involves chemical insecticides, but overuse has brought about many negative effects. Our previous study reported that (S)-(−)-palasonin (PLN) is a plant-derived active substance with significant insecticidal activity against *P. xylostella*. However, we noticed a possible cross-resistance between (S)-(−)-palasonin and other insecticides which may be related to metabolic detoxification. In order to further explore the detoxification effect of detoxification enzymes on (S)-(−)-palasonin in *P. xylostella*, the effects of (S)-(−)-palasonin on enzyme activity and transcription level were determined, and the detoxification and metabolism of GSTs on (S)-(−)-palasonin were studied by in vitro inhibition and metabolism experiments. During this study, GST enzyme activity was significantly increased in *P. xylostella* after (S)-(−)-palasonin treatment. The expression levels of 19 GSTs genes were significantly increased whereas the expression levels of 1 gene decreased. Furthermore, (S)-(−)-palasonin is shown to be stabilized with GSTs and metabolized GSTs (GSTd1, GSTd2, GSTs1 and GSTs2) in vitro, with the highest metabolic rate of 80.59% for GSTs1. This study advances the beneficial utilization of (S)-(−)-palasonin as a botanical pesticide to control *P. xylostella* in the field.

## 1. Introduction

*Plutella xylostella* (L.) is a worldwide pest [1,2,3]. Currently, the management of *P. xylostella* primarily depends on the usage of chemical insecticides [4,5]. The classes of insecticides widely used against *P. xylostella* include pyrethroids, organochlorines, insect growth regulators, carbamates, neonicotinoids, organophosphates, etc. [6,7,8]. However, a substantial number of studies have shown that *P. xylostella* has strong adaptability to various insecticides, rapid development of resistance, and serious cross-resistance to most commonly used insecticides [9,10]. Therefore, it is necessary to find a new effective insecticide to control *P. xylostella* and delay the development of resistance.

Our previous studies have found that (S)-(−)-palasonin which is extracted from *Butea monosperma* seeds has great application value in the management of *P. xylostella*. (S)-(−)-palasonin has strong ingestion toxicity and contact toxicity to *P. xylostella.* The LC_50_ of ingestion toxicity and LD_50_ of contact toxicity at 48 h is 10.72 mg/L and 0.22 µg/larva, respectively. In addition, (S)-(−)-palasonin also has an insecticidal effect on the field strain of *P. xylostella*. (S)-(−)-palasonin has the potential to become a pesticide resistance management agent of *P. xylostella*, which can be used in a comprehensive management strategy. However, we noticed that field strains without exposure to (S)-(−)-palasonin still produced low levels of resistance, with a resistance ratio of 9.47-fold [11]. This suggests that there may be some cross-resistance between (S)-(−)-palasonin and other insecticides. The cause of cross-resistance may be related to metabolic resistance or target resistance [12,13] and we presume that metabolic mechanisms may play a more important function in (S)-(−)-palasonin resistance than target site resistance. Insect resistance to insecticides is mainly through two mechanisms: reducing the sensitivity of target enzymes and increasing the metabolic capacity of detoxification enzymes [14]. Li et al. found that insects do not easily develop resistance to cantharidin and its analogues through target enzymes. The structure of (S)-(−)-palasonin and cantharidin is similar and the key target enzymes for *P. xylostella* are the same. We presume that detoxification enzyme-mediated metabolism may be one of the reasons for the resistance of *P. xylostella* to (S)-(−)-palasonin.

Metabolic regulation mediated by insect detoxification enzymes is one of the important mechanisms of insecticide resistance [15,16]. Insects resist the invasion of foreign toxic substances by increasing the activity of detoxification enzymes or increasing the metabolic capacity of detoxification enzymes, mainly including: carboxylesterase (CarEs), cytochrome P450 (P450) and insect glutathione S-transferase (GSTs). These insect detoxification enzymes can be individually or synergistically implicated in the detoxification metabolism of different insecticides [17]. Yang et al. found that the activities of glutathione S-transferases (GSTs), carboxylesterase (CarE) and acetylcholinesterase (AChE) in the resistant strain were higher than those in the susceptible strain [18].

In this study, the detoxification effect of the field strain of *P. xylostella* on (S)-(−)-palasonin was evaluated by comparing the changes in detoxifying enzyme activity. The detoxification enzyme responses were characterized after (S)-(−)-palasonin treatment. Additionally, the expressions of GST genes altered following (S)-(−)-palasonin exposure and the role of GSTs in the detoxification process of (S)-(−)-palasonin were further validated. Our study provides a further theoretical basis for the detoxification of xenobiotics by *P. xylostella* and the exploitation of (S)-(−)-palasonin.

## 2. Materials and Methods

### 2.1. Plutella xylostella

The sensitive strain of *Plutella xylostella* was supplied by the Key Lab of Plant Protection Resources & Pest Management of Ministry of Education (Northwest A&F University, Yangling, China). *P. xylostella* was kept indoors at 25 ± 2 °C, RH 50 ± 5% and photoperiod 16 h:8 h (light:dark) for 10 years continuously without exposure to any pesticides [19]. The source of the field strain of *P. xylostella* is the same as Fan et al. The larvae fed on *Brassica oleracea* leaves, and the adults fed on 10% honey solution.

### 2.2. (S)-(−)-Palasonin Exposed

(S)-(−)-palasonin (PLN, CAS: 11043-72-4, purity > 95%) was extracted and purified from the seeds of *Butea monosperma* (Lam.) by the method of Fan et al. [11]. The LC_10_ (6.98 mg/L), LC_25_ (8.65 mg/L), and LC_50_ (10.72 mg/L) values of (S)-(−)-palasonin against the 3rd instar larvae of *P. xylostella* were calculated according to the toxicity regression equation (y = −12.15 + 11.80x, χ^2^ = 2.20, *p* = 1.00) of 48 h treated with (S)-(−)-palasonin.

In enzyme activity tests, 3rd instar larvae of *P. xylostella* were treated with LC_10_ and LC_25_ concentrations of (S)-(−)-palasonin by the leaf dipping method [11]. Each treatment was inoculated in 300 larvae. A treatment with a 20% acetone–water mixture was used for the control group. After treatment, 20 test insects were collected at 3 h, 6 h, 12 h, 24 h and 48 h after treatment, respectively, and were rapidly frozen in liquid nitrogen and kept at −80 °C before enzyme activity analyses.

Larvae (3rd instar) were treated with LC_50_ concentrations of (S)-(−)-palasonin by the aforementioned leaf-dipping method for the GSTs transcriptional level analysis. Larvae samples were collected at 6 h after treatment. Each replicate contained twenty larvae. Three parallels were made for each concentration.

### 2.3. Enzyme Activity Assay

Detoxification enzymes (GSTs, P450 and CarE) activities were determined with reference to the method of Li et al. [20]. For GST, the working solution included 2 μL of enzyme, 2 μL of GSH, 10 μL of 1-chloro-2,4-dinitrobenzene (CDNB, Solarbio, Beijing, China) and 186 μL sodium phosphate buffer. Then, the absorbance at 340 nm every 1 min for 6 times was measured. For CarE, the working solution was incubated at 25 °C for 15 min, which included 5 μL of enzyme, 190 μL of Tris-HCl/CaCl_2_ (25 mM/1 mM, pH 7.0) and 5 μL of α-NA (0.1 mM). Then, 25 μL fast blue B salt (0.2%) (MP Biochemicals, Irvine, CA, USA) and 25 μL sodium dodecylsulfate (SDS, 0.42%) were added to the mixture and incubated at room temperature for 30 min under dark. The absorbance was recorded at 600 nm. For P450, the working solution was incubated at 30 °C for 2 h, which included 10 μL of enzyme, 10 μL of nicotinamide adenine dinucleotide phosphate (NADPH, 9.6 mM), 2 mM of p-nitroanisole and 180 μL sodium phosphate buffer (100 mM with 0.5% Triton X-100, pH 7.2). The absorbance was recorded at 405 nm.

The relative inhibition, relative activity and depletion of (S)-(−)-palasonin were calculated as follows:(1)$$\mathrm{Relative}\;\mathrm{activity}\;\mathrm{level}=\frac{\mathrm{Field}\;\mathrm{strain}\;\mathrm{enzyme}\;\mathrm{activity}}{\mathrm{Sensitive}\;\mathrm{strain}\;\mathrm{enzyme}\;\mathrm{activity}}$$
(2)$$\mathrm{Relative}\;\mathrm{activity}\;\mathrm{level}=\frac{\mathrm{Slope}\;\mathrm{of}\;\mathrm{CT}}{\mathrm{Slope}\;\mathrm{of}\;\mathrm{NC}}$$

In formula, NC represents the GST enzyme control without (S)-(−)-palasonin and CT represents the sample test with inhibitor ((S)-(−)-palasonin). All assays were replicated 3 times.

### 2.4. Quantitative Reverse Transcription Polymerase Chain Reaction (qRT-PCR)

To identify the expression levels of *GSTd1* (GenBank: KF929200), *GSTd2* (GenBank: KF929201), *GSTd3* (GenBank: AB541016), *GSTd4* (GenBank: KF929202), *GSTd5* (GenBank: XM011560668), *GSTe1* (GenBank: KF929203), *GSTe2* (GenBank: KF929204), *GSTe3* (GenBank: U66342), *GSTe4* (GenBank: KF929205), *GSTe5* (GenBank: KF929206), *GSTe6* (GenBank: KF929207), *GSTo1* (GenBank: KF929208), *GSTo2* (GenBank: KF929209), *GSTo3* (GenBank: KF929210), *GSTs1* (GenBank: AB180447), *GSTs2* (GenBank: AB180454), *GSTt1* (GenBank: KF929211), GSTu1 (GenBank: KF929213), *GSTu2* (GenBank: KF929214) and *GSTz1* (GenBank: KF929212) responding to different times of (S)-(−)-palasonin exposure. qRT-PCR was performed with the primers in Appendix A. Primers were designed using the Integrated DNA Technologies (IDT) primer quest tool (https://sg.idtdna.com/PrimerQuest/Home/Index (accessed on 6 December 2021). The gene *β-actin* (GenBank: JN410820) was used as the internal reference gene.

The qRT-PCR was conducted using SYBR Premix Ex Taq II (Tli RNase H Plus) (Takara, Dalian, China) on a Light Cycler^®^ 480 (Roche, Basel, Switzerland). The 20 μL reaction system consisted of 1 μL cDNA template, 0.8 μL forward gene-specific primers, 0.8 μL reverse gene-specific primers, 10 μL SYBR Premix Ex Taq and 6.9 μL of double-distilled water. We applied the following 2-step method: 95 °C for 30 s, 45 cycles including of 94 °C for 5 s and 60 °C for 30 s. Melting curves were auto-generated to verify the specificity of the PCR product. All qRT-PCR analyses were performed in 3 biological replicates with 3 technology replicates. The 2^−ΔΔCt^ method was applied to calculate the relative expression of GST genes [21].

### 2.5. Expression of the Plutella xylostella Recombinant GST Proteins (GSTs)

The expression and purification of GSTs protein were performed by the previously published methods [22]. The GSTs genes were incorporated into the pET-30a (+) vector (TaKaRa, Dalian, China). We transformed *E. coli* BL21 (DE3) cells (Weidi, Shanghai, China) with the recombinant plasmid and incubated them in an LB medium containing kanamycin (100 μg/mL) until the log phase (OD600 = 0.6). Isopropyl-β-D-thiogalactoside (IPTG, 0.1 mM) was then added to the bacterial broth and incubated at 200 rpm, 30 °C for 6 h. The bacteria were pelleted by centrifugation (8000× *g*, 4 °C), then resuspended in PBS (10 mM, pH 7.4). After sonication, the lysate was centrifuged and the target proteins purified from the supernatant by Ni^2+^-NTA affinity column (Smart-Lifesciences, Changzhou, China), verified by SDS-PAGE and dialyzed overnight. The protein concentration was determined by BCA, and the obtained GSTs proteins were preserved at −80 °C for subsequent experiments.

### 2.6. Enzymatic Activity and Inhibition Assays

The enzymatic activity assay for recombinant GSTs protein was improved by using 1-chloro-2,4-dinitrobenzene (CDNB) as a substrate and referring to the method of Li et al. [20]. Different concentrations of CDNB (0.05–1.6 mM), 1 μg protein and GSH diluent (1 mM, dissolved in 100 mM phosphate buffer) were pipetted in the clear 96-well plates (ThermoFisher, Waltham, MA, USA). The absorbance was measured every 1 min for 6 times at 340 nm using an M200 Microplate Reader (Tecan, Männedorf, Switzerland).

Inhibition assays were carried out in the same manner as the above enzyme activity assay test. Serial dilutions of (S)-(−)-palasonin were mixed with 1 μL CDNB (1 mM), 1 μg recombinant protein and 100 μL PBS (100 mM) at 25 °C for 10 min prior to the addition of the substrate GSH. The reaction was then initiated by sequentially adding 1 μL of GSH (1 mM) and PBS (100 mM), resulting in a total volume of 200 μL of the reaction system.

The relative inhibition was calculated as follows:(3)$$\mathrm{Relative}\;\mathrm{inhibition}\;\mathrm{level}=\frac{\mathrm{Slope}\;\mathrm{of}\;\mathrm{NC}-\mathrm{Slope}\;\mathrm{of}\;\mathrm{CT}}{\mathrm{Slope}\;\mathrm{of}\;\mathrm{NC}}$$

In this formula, NC represents the peak area of the control group and CT represents the peak area of (S)-(−)-palasonin treatment group.

### 2.7. Metabolism Assays

The metabolic assay was performed by gas chromatography (GC) (Shimadzu, Japan). (S)-(−)-palasonin (100 mM) was solubilized in DMSO as a reserve solution. The reserve solution was dissolved in sodium phosphate buffer (PBS) (50 mM, pH 7.4) to achieve a 0.5 mM workup solution for subsequent experiments. A total of 1 mL of (S)-(−)-palasonin working solution was incubated with 1 mL of recombinant GSTs protein dilution (containing 50 μg of recombinant protein) at 30 °C for 10 min, then added a final concentration of 5 mM GSH (solubilized in PBS buffer as previously described). Incubated the above mixture for 3 h at 30 °C with shaking at 200 rpm. Subsequently 500 μL dichloromethane aliquots were added to terminate the reaction. Transfer the dichloromethane phase to sample vials for GC detection. GC analytical conditions refer to the method reported by Fan et al.; the primary temperature was maintained at 60 °C for 2 min, then increased to 320 °C at 8 °C/min and held for 10 min. The temperature of the sampler and detector was set at 330 °C [11].

The depletion of (S)-(−)-palasonin was calculated as follows:(4)$$\mathrm{Depletion}\;\mathrm{of}\;\mathrm{PLN}\left(\%\right)=\frac{\mathrm{Peak}\;\mathrm{area}\;\mathrm{of}\;\mathrm{NC}-\mathrm{Peak}\;\mathrm{area}\;\mathrm{of}\;\mathrm{CT}}{\mathrm{Peak}\;\mathrm{area}\;\mathrm{of}\;\mathrm{NC}}\times 100$$

In this formula, NC represents the GST enzyme control without (S)-(−)-palasonin and CT represent the sample test with inhibitor ((S)-(−)-palasonin).

### 2.8. Statistical Analysis

The significance of this data was analyzed by SPSS 20.0 and the one-way analysis of variance (ANOVA) was used for multiple sets of data.

## 3. Results and Discussion

### 3.1. Activity Changes of Detoxifying Enzymes in Field Strain of Plutella xylostella

Based on the results from our previous study, we found that the field strain of *P. xylostella*, which was never exposed to (S)-(−)-palasonin, produced a low level of resistance to (S)-(−)-palasonin by 9.47-fold [11]. It showed that (S)-(−)-palasonin has a certain cross-resistance with other insecticides. This resistance may be closely related to metabolic resistance mediated by detoxification enzymes. By measuring the activity of crude detoxification enzymes in the field strain of *P. xylostella*, GST and P450 enzyme activities were found to be significantly higher in sensitive strains, which could reach 3.17- and 1.82-fold, respectively (Figure 1). We speculate that glutathione S-transferase has a closer detoxification effect on (S)-(−)-palasonin in *P. xylostella*.

It has been shown that the increased activity of insect detoxification enzymes is implicated in the formation of insecticide resistance [23,24,25]. Ruttanaphan et al. pointed out that the activities of P450, GST, and CarE in cypermethrin resistance populations of *Spodoptera litura* were significantly increased when compared to susceptible populations [26]. Over-expression of P450 genes is the cause of a 109-fold resistance to permethrin in *Musca domestica* [27]. In *Tetranychus cinnabarinus*, the increased GST activity and overexpression of *GSTs* are thought to be responsible for a 104.07-fold resistance to cyflumetofen [28]. These studies all suggest that the increased activity of insect detoxification enzymes is related to insecticide resistance. The presence of significantly increased GST and P450 enzyme activities in the present study is observed in *P. xylostella*, especially where GST showed the highest changes, i.e., 3.17-fold of the GST activity in the susceptible strain. Thus, GST may be the primary detoxifying enzyme for the resistance to (S)-(−)-palasonin in *P. xylostella*.

### 3.2. GST Activity

In order to further detect the stress response of GST to (S)-(−)-palasonin in *P. xylostella*, we detected the change of GST activity in *P. xylostella* treated with LC_10_ and LC_25_ concentrations of (S)-(−)-palasonin for 48 h (Figure 2). The results show that GST enzymatic activity shows a decreasing trend followed by an increasing trend at both concentrations. The GST activity was lower after 3–12 h of treatment in comparison with the control group, and the enzyme activity attained its lowest value after 6 h of treatment, which was 0.56 (LC_10_ concentration) and 0.42 (LC_25_ concentration) of the control group, respectively. Nevertheless, the GST activity of LC_10_ and LC_25_ treatment groups increased continuously after 24 h. The treatment group increased 1.47-fold and 1.54-fold, respectively, compared with the control group after 48 h.

Numerous studies indicate that the increase in GST activity is related to the enhancement of the detoxification ability of insects to insecticides [29,30,31]. Farouk et al. reported that the increase in GST activity was correlated with the detoxification of the insecticides malathion and permethrin by *P. xylostella* [32]. When exposed to multiple insecticides, GST activity was significantly greater in wild-type populations of *Panonychus citri* than in sensitive populations, and this increase in GST was strongly related to the intensity of detoxification of *Panonychus citri* to insecticides [14]. These studies showed that the enhancement of detoxification enzyme activity may be related to insect sensitivity to insecticides. In this study, the GST activity of *P. xylostella* exposed to (S)-(−)-palasonin for 24 h was dramatically above that of the control. The increasing trend of GST activity in response to (S)-(−)-palasonin is consistent with some other research results, indicating that GST is probably involved in the detoxification of (S)-(−)-palasonin by *P. xylostella*.

### 3.3. Expression Levels Effect of GSTs Exposure to (S)-(−)-Palasonin

There are 22 cytoplasmic GSTs identified in the genome of *P. xylostella*, divided into six subgroups: Omega, Sigma, Theta, Zeta, Delta and Epsilon [33,34]. To further determine whether the GSTs of *P. xylostella* are involved in the detoxification of (S)-(−)-palasonin, the transcription of 20 GST genes of *P. xylostella* was studied under LC_50_ concentration of (S)-(−)-palasonin exposure. The relative gene expression levels of (S)-(−)-palasonin-treated larvae and controls were compared by qPCR (Figure 3). The results show that the transcriptional level of seven GSTs genes increased first and then decreased after treatment with (S)-(−)-palasonin. The transcriptional level of three GSTs genes decreased first and then increased and the transcriptional level of five GSTs genes decreased gradually, the transcriptional level of five GSTs genes decreased first and then increased and decreased again after 24 h. However, except for GSTo3, most GSTs were up-regulated at different times after (S)-(−)-palasonin treatment.

Previous studies have shown that exogenous substances can induce GST gene upregulation in insects and this suggests GST is involved in insecticide detoxification. In *Culex pipiens*, *CpGSTd1* was reported to be involved in 1,1,1-trichloro-2,2-bis (4-chlorophenyl) ethane (DDT) detoxification [25]. Furthermore, pyrethroid induced a higher expression of the Nlgst1-1 in the pyrethroid-resistant strain of *Nilaparvata lugens* [35]. After silencing the *NlGSTe1*, the sensitivity of nymphs to chlorpyrifos increased significantly in *Nilaparvata lugens* [36]. In this study, our findings indicate that upon contact with (S)-(−)-palasonin, *P. xylostella* larvae rapidly responded to this adversity by upregulating the expression levels of GSTs engaged in defense against (S)-(−)-palasonin. Almost all GSTs have an up-regulation trend at different time points ranging from 1.67 to 4.07-fold, and this difference may be due to the different stress responses of different genes to (S)-(−)-palasonin. We speculate that these genes may be related to the detoxification metabolism of (S)-(−)-palasonin, but it is impossible to determine which genes are more directly related to the detoxification effect of *P. xylostella* on (S)-(−)-palasonin.

### 3.4. Recombinant Protein Expression and Kinetics of GSTs

Not all GST proteins exhibit binding activity to CDNB [37]. Of the published *P. xylostella* GSTs, 10 recombinant GSTs proteins had both GSH and H sites, including GSTd1, GSTd2, GSTd3, GSTd4, GSTd4, GSTd5, GSTe4, GSTe5, GSTs1, GSTs2 and GSTu1 [34].

In this study, these nine recombinant GSTs (GSTd1, GSTd2, GSTd3, GSTd4, GSTd4, GSTd5, GSTe4, GSTe5, GSTs1 and GSTs2) proteins were accessed by a prokaryotic expression system and the recombinant proteins were expected to have a molecular weight of 25 k Da (Appendix A). The yield of nine purified proteins was 0.6 μg/μL, 1.8 μg/μL, 0.33 µg/µL, 0.95 µg/µL, 0.7 µg/µL, 1.22 µg/µL, 0.62 µg/µL, 1.43 μg/μL and 0.48 μg/μL, respectively.

The values of K_m_ of GSTd1, GSTd2, GSTd3, GSTd4, GSTd4, GSTd5, GSTe4, GSTe5, GSTs1 and GSTs2 were 0.6908 mmol/L, 0.6821 mmol/L, 1.006 mmol/L, 1.121 mmol/L, 0.5458 mmol/L, 0.4132 mmol/L, 0.7048 mmol/L, 0.6844 mmol/L and 0.4689 mmol/L, respectively. The values of V_max_ (µmol/min/mg) were 375.8, 3177, 2013, 690.9, 726.1, 1215, 520.2, 1682 and 362.3, respectively.

Previous reports showed that the value of K_m_ of *P. xylostella* recombinant GSTd could reach 241.7 ± 3.25 µmol/min/mg [22]. The kinetic results of recombinant GSTs show that the recombinant GSTs was better than the previously reported results. Therefore, the purified recombinant GSTs can be tested as follows (Appendix A).

### 3.5. Inhibitory and Metabolism Effects between (S)-(−)-Palasonin and GSTs

The inhibition of GST by insecticides can be reflective of the intimate affinity of GST for insecticides and is considered as having relevance to the detoxification and metabolism of insecticides [22,38]. The inhibitory effects of (S)-(−)-palasonin on GSTs were determined (Figure 4). The results show that the activities of GSTd1, GSTd2, GSTs1 and GSTs2 proteins in recombinant *P. xylostella* are significantly inhibited by (S)-(−)-palasonin at 1 mM concentrations, which are 0.35, 0.55, 0.26 and 0.51 of the control values, respectively. Nevertheless, at the concentration of 0.1 mM, (S)-(−)-palasonin has little inhibitory effect on GSTs (Figure 4). The metabolic function of insect GST is one of the important mechanisms for insects to resist endogenous and exogenous toxins [23,24]. This study examined the metabolic effect of purified GSTs proteins on (S)-(−)-palasonin using the GC method (Figure 5). The results demonstrate that GSTd1, GSTd2 and GSTs1 metabolized 78.33%, 79.37% and 80.59% of (S)-(−)-palasonin in the 3-h reaction. Additionally, GSTs2 metabolized 37.30% (S)-(−)-palasonin, which is close to half of that of GSTd1 (Figure 5).

GST has a variety of detoxification and metabolic methods for harmful substances. First, it can improve its water solubility by catalyzing the combination of toxic substances with thiol groups on reduced glutathione (GSH) to facilitate its excretion from insects [39,40]. Liu et al. [41] reported that CpGSTd1 was inhibited by lambda-cyhalothrin in *Cydia pomonella*, and CpGSTd1 had significant metabolic interaction to lambda-cyhalothrin, which might be one of the reasons for the resistance of the worms to lambda-cyhalothrin. Concomitantly, some studies have also found that some GSTs proteins only bind to insecticides, but do not participate in the degradation of insecticides. Kostaropoulos et al. [42] found that deltamethrin in *Tenebrio molitor* inhibited the binding of GST to the active site of CDNB in a competitive manner but did not bind to GSH. On the other hand, GST can directly metabolize toxic substances into other nontoxic substances [42,43] as in the case where GSTs of *Liposcelis entomophila* had the capacity of metabolizing malathion, propuxor and deltamethrin [44].

In this study, the recombinant *P. xylostella* GSTd1, GSTd2, GSTs1 and GSTs2 are inhibited in vitro, indicating that the affinity of (S)-(−)-palasonin to GSTs is superior to that of substrate CDNB, thereby inhibiting the binding reaction of CDNB with GSH mediated by GSTs. The potent binding ability of (S)-(−)-palasonin to GSTs is assumed to be critical evidence that GSTs are accountable for (S)-(−)-palasonin detoxification. Moreover, in vitro experiments show that GST can not only be inhibited by (S)-(−)-palasonin but also effectively metabolize (S)-(−)-palasonin. More specifically, (S)-(−)-palasonin has a high inhibitory activity on GSTs1, which efficiently metabolized (S)-(−)-palasonin.

(S)-(−)-palasonin is a chiral compound, often associated with a nitrogenous acidic compound in seeds [45]. In other words, (S)-(−)-palasonin exhibits poor chemical and structural stability. Customarily, the better the chemical stability, the lower the metabolic hydrolysis rate. Battershill et al. found that the water solubility of compounds is significantly improved by introducing groups to convert symmetrical molecular structure compounds into asymmetric structures [46]. We speculate the high metabolic rate of recombinant GSTs to (S)-(−)-palasonin may be caused by the structural instability of (S)-(−)-palasonin.

## 4. Conclusions

Overall, the detoxification of (S)-(−)-palasonin by GST may be mainly through direct metabolism in *P. xylostella*. The above findings demonstrate that GSTd1, GSTd2, GSTs1 and GSTs2 are implicated in the detoxification process of (S)-(−)-palasonin in *P. xylostella*. Among the four GSTs, GSTs1 had the strongest metabolizing ability towards (S)-(−)-palasonin. This study contributes to the understanding of the mechanism underlying resistance to (S)-(−)-palasonin in *P. xylostella* and provides a theoretical basis for the application of (S)-(−)-palasonin in *P. xylostella* field control.

## Figures and Tables

**Figure 1 insects-13-00989-f001:**
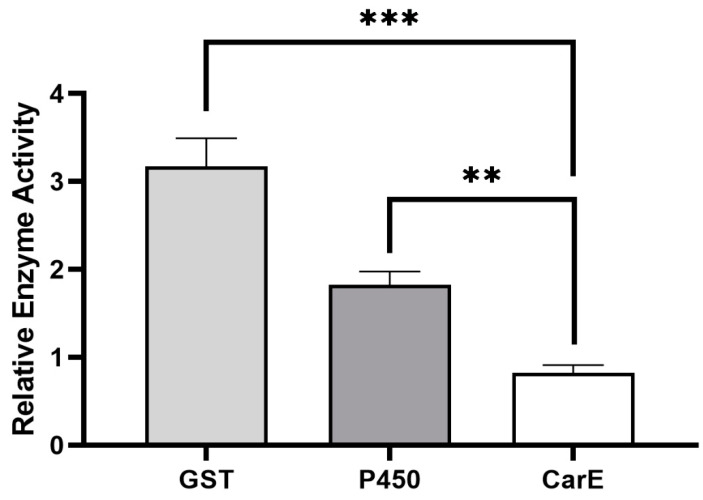
Differences of detoxification enzyme activities between field strains and susceptible strains of *Plutella xylostella*. (Bars indicate means ± standard errors of replicates. Asterisks indicate statistically significant variances (one-way ANOVA and Tukey’s multiple comparison test were used; ***, *p* ≤ 0.001; **, *p* ≤ 0.01.).

**Figure 2 insects-13-00989-f002:**
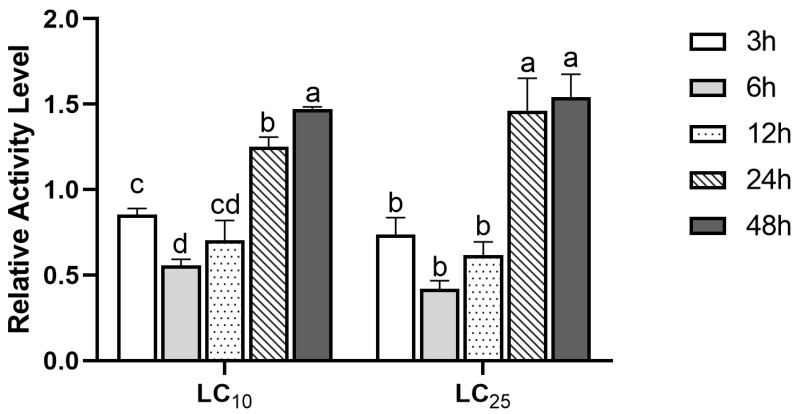
Effects of LC_10_ and LC_25_ concentrations of (S)-(−)-palasonin on the GST activities in *Plutella xylostella* (Bars indicate means ± standard errors of replicates. Different alphabets above indicate significant variances (*p* < 0.05) calculated with one-way ANOVA, followed by Tukey’s multiple comparison test.).

**Figure 3 insects-13-00989-f003:**
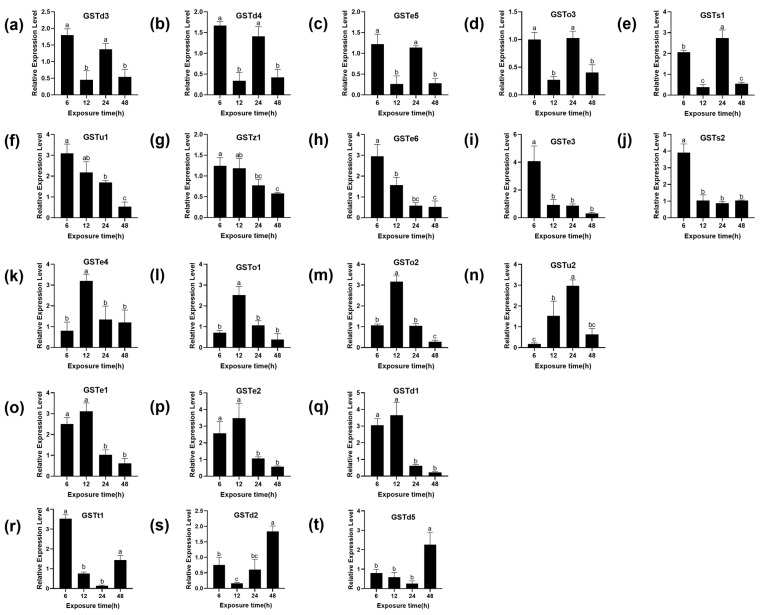
Effects of (S)-(−)-palasonin on the expression level of GSTs genes in *Plutella xylostella*. GSTd3 (**a**), GSTd4 (**b**), GSTe5(**c**), GSTo3 (**d**), GSTs1 (**e**), GSTu1 (**f**), GSTz1 (**g**), GSTe6 (**h**), GSTe3 (**i**), GSTs2 (**j**), GSTe4 (**k**), GSTo1 (**l**), GSTo2 (**m**), GSTu2 (**n**), GSTe1 (**o**), GSTe2 (**p**), GSTd1 (**q**), GSTt1 (**r**), GSTd2 (**s**) and GSTd5 (**t**). (Bars indicate means ± standard errors of replicates. Different alphabets above indicate significant variances (*p* < 0.05) calculated with one-way ANOVA, followed by Tukey’s multiple comparison test.).

**Figure 4 insects-13-00989-f004:**
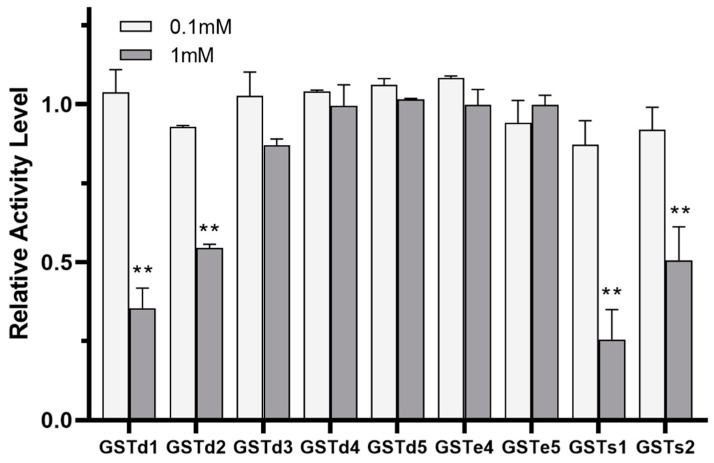
Inhibition of (S)-(−)-palasonin on GSTs protein. (Bars indicate means ± standard errors of replicates. Asterisks indicate statistically significant variances (one-way ANOVA and Tukey’s multiple comparison test were used; **, *p* ≤ 0.01.).

**Figure 5 insects-13-00989-f005:**
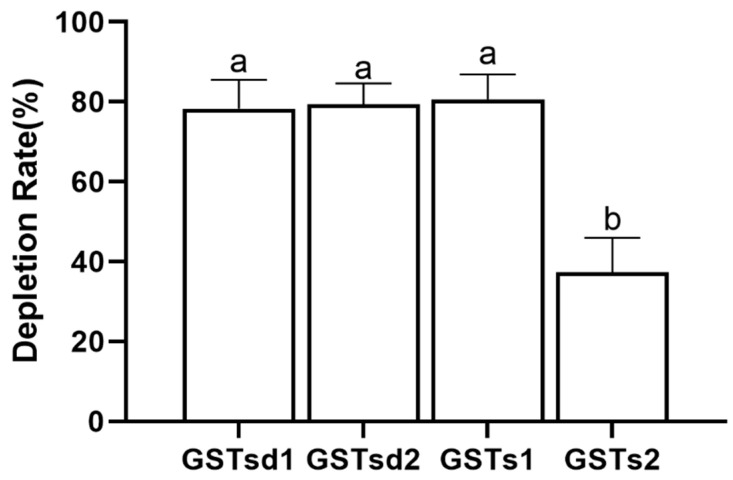
Metabolic capacity of four GSTs proteins toward (S)-(−)-palasonin. (Bars indicate means ± standard errors of replicates. Different alphabets above indicate significant variances (*p* < 0.05) calculated with one-way ANOVA, followed by Tukey’s multiple comparison test.).

## Data Availability

Data available on request.

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
