# Peer review of "Glutathione S-Transferase May Contribute to the Detoxification of (S)-(−)-Palasonin in Plutella xylostella (L.) via Direct Metabolism"

_insects, 2022, doi:10.3390/insects13110989_

Round 1

Reviewer 1 Report

1. Line 206: The word increase should be in the past tense “increased”

2. Why do you choose to use only one housekeeping gene (B-actin) as an internal control for the qPCR? The acceptable and reliable way is always to use two or more reference genes.

3. I am still wondering why would you use enzyme activity tests in some cases and qPCR in others. You need to explain this in the discussion detailing why each is appropriate where it was used.

4. In this paper, all the GSTs were from  Plutella xylostella, I do not think it is necessary to write “Px” before each GST. Removing Px will make it easier for readers, especially when you have many of them written.

Author Response

Dear Editors and Reviewers:

We are very appreciative of your suggestions on the manuscript entitled “Glutathione S-transferase may contribute to the detoxification of (S)-(-)-palasonin in Plutella xylostella (L.) via direct metabolism” (ID: insects-1968948). We have studied the comments carefully and made corrections which we hope to meet with approval. Revised portions are marked in red on the paper. The main corrections in the paper and the responses to the reviewer’s comments are as follows:

Reviewer #1:
Comments and Suggestions for Authors:

  1. Line 206: The word increase should be in the past tense “increased”

Response: We thank the reviewer for pointing out these errors, which we have now corrected.

  1. Why do you choose to use only one housekeeping gene (β-actin) as an internal control for the qPCR? The acceptable and reliable way is always to use two or more reference genes.

Response: Thank you for your very helpful suggestions to improve our paper. We agree with your suggestion. But there are precedents for the β-actin from P. xylostella (GenBank JN410820) as the only internal reference genes, such as Bahar et al., 2013; Shi et al., 2014; Zhang, 2015.

Bahar, M. H.; Hegedus, D.; Soroka, J.; Coutu, C.; Bekkaoui, D.; Dosdall, L. Survival and Hsp70 gene expression in Plutella xylostella and its larval parasitoid Diadegma insulare varied between slowly ramping and abrupt extreme temperature regimes. PLoS One, 2013, 8(9), e73901.

Shi, M.; Chen, X. Y.; Zhu, N.; Chen, X. X. Molecular identification of two prophenoloxidase-activating proteases from the hemocytes of Plutella xylostella (Lepidoptera: Plutellidae) and their transcript abundance changes in response to microbial challenges. Journal of Insect Science, 2014, 14(1), 179.

Zhang, Y. Identification and characterization of NADPH-dependent cytochrome P450 reductase gene and cytochrome b5 gene from Plutella xylostella: Possible involvement in resistance to beta-cypermethrin. Gene, 2015, 558(2), 208-214.

  1. I am still wondering why would you use enzyme activity tests in some cases and qPCR in others. You need to explain this in the discussion detailing why each is appropriate where it was used.

Response: Thank you for this comment, and we are sorry for not being clear enough. (S)-(-)-palasonin is a plant-derived compound with great application value in the control and management of P. xylostella. Our previous study have found that (S)-(-)-palasonin may has cross-resistance with other insecticides which may be related to metabolic detoxification metabolism (Fan et al., 2022). However, it is impossible to determine which detoxification enzymes are more effective in the detoxification of (S)-(-)-palasonin in P. xylostella. Insects detoxification enzymes mainly including: carboxylesterase (CarEs), cytochrome P450 (P450) and insect glutathione S-transferase (GSTs) (Guo et al., 2014; Xin and Zhang, 2021). By measuring the activity of crude detoxification enzymes (GSTs, CarEs, and P450) in the field strain of P. xylostella, we found that GST activity changes most significantly compared to other detoxification enzymes. Therefore, we chose GST as the main research object to further study including qPCR, metabolism and inhibition test in vitro.

Fan, Q.; Li, X.; Wei, C.; Wang, P.; Sun, H.; Zheng, S.; Li, Y.; Tian, Z.; Liu, J.; Zhang, Y. Extraction, structure characterization and biological activity determination of (S)-(-)-palasonin from Butea monosperma (Lam.) Kuntze seeds. Industrial Crops and Products, 2022, 187.

Guo, L.; Xie, W.; Wang, S.; Wu, Q.; Li, R.; Yang, N.; Yang, X.; Pan H.; Zhang, Y. Detoxification enzymes of Bemisia tabaci B and Q: biochemical characteristics and gene expression profiles. Pest management science, 2014, 70(10), 1588-1594.

Xin, S.; Zhang, W. Construction and analysis of the protein–protein interaction network for the detoxification enzymes of the silkworm, Bombyx mori. Archives of Insect Biochemistry and Physiology, 2021, 108(4), e21850.

  1. In this paper, all the GSTs were fromPlutella xylostella, I do not think it is necessary to write “Px” before each GST. Removing Px will make it easier for readers, especially when you have many of them written.

Response: Thank you for your valuable comments on our paper which we have amended according to your suggestions. Once again, thank you very much for your comments and suggestions.

Reviewer 2 Report

This manuscript does a very nice job of detailing how GSTs might be involved in the detoxification of (S)-(-)-palasonin in Plutella xylostella.  The data clearly point to differences between the sensitive and non-sensitive Plutella xylostella, changes in GST activity following exposure to sub lethal amounts of (S)-(-)-palasonin, and changes in the expression of the GSTs.  Further the recombinant GST proteins indicate that GSTd1, GSTd2, GSTs1, and GSTs2 are actively involved in the metabolism.  On the whole, i think the work is well done, it appears time and care was put into the preparation of the manuscript, and finally the figures are simple and easy to interpret.  A few points that will improve the manuscript is the incluse of the methods for the detoxification enzymes (GST, P450, and CarE) in the methods section, correcting the formula used to calculate PLN metabolism, and a consideration of rearranging of figure 3 to group the data by expression profile instead of numerical order.

Items to improve this work.  

Line 12:  “has a good toxic” Good is a very broad qualifier, so maybe saying “competent” better fits your meaning.

23 & 35:    PLN - this term is not previously defined, therefore it needs to be defined or spelled out.

53:  The term “stomach” should be replaced by midgut.

54:   LC50 is in units of mg/L, you should replace that value with molarity, or at minimum include molarity in parenthesis afterward.

101:  References a “leaf dipping method above” but it is not described above.

112:  You reference the methods for GST, P450 and CarE in Li et al. You need to follow that statement with a brief description (this serves as a method for a reader who may not have access to that article the ability to understand how the data was collected). 

149-151:  Bacterial cells are commonly described as cell “pellet” instead of cell “precipitate”.  I’m thinking this should read, The bacteria were pelleted by centrifugation (8000 g, 4ËšC), then resuspended in PBS.  After sonication the lysate was centrifuged and the target proteins purified from the supernatant by Ni-NTA.  Or something similar.

157:  The abbreviation “CDNB” is used but is not defined.  Please use the molecules full name followed by the abbreviation in parenthesis.

159:  not “instilled” were pipetted.

179:  “dichloromethane mixtures”  should read dichloromethane “aliquots” were added….

185:  The formula for the calculation is incorrect.  As written “In this formula, NC represents the GST enzyme control without (S)-(-)-palasonin and CT represents the sample test with inhibitor ((S)-(-)-palasonin.”  If this were correct the peak area without PLN should be near zero, then subtracting the peak area of the sample with PLN would result in a negative value, which is then divided by a near zero number.  This shouldn’t be. I’m assuming that in the experiment NC is without PxGST, and CT is with PxGST.  If this is not the case a much better explanation of the experiment is required, and I cannot accept that experiment as valid.

236: “this increment” is awkward,  “this increase” makes more sense.

274:  I appreciate the numerical order to the plots, but I think it would be better if the order were to group the expression patterns.  For example, d3, d4, e5, o3, s1 in a single column or row because they are all up at 6 and 24 hours while down at 12 and 48.

284: …molecular weight of c. 25 kD, remove the c.

284: The yield of four, but you list 9 proteins.

285: c. need to be removed again

290: Since all the units for the vmax are the same, it might be less distracting to say, the values for vmax (in μmol/min/mg) are….

300:  “as have” should be “as having”

336:  you use icorr, I would write it out

Author Response

Dear Editors and Reviewers:

Thank you for giving us an opportunity to revise our manuscript, we appreciate editors and reviewers very much for their positive and constructive comments and suggestions on our manuscript entitled “Glutathione S-transferase may contribute to the detoxification of (S)-(-)-palasonin in Plutella xylostella (L.) via direct metabolism” (ID: insects-1968948).Those comments are all valuable and very helpful for revising and improving our paper, as well as the important guiding significance to our researches. We have studied comments carefully and have made correction which we hope meet with approval. The main corrections in the paper and the responds to the reviewer’s comments are as follows:

Reviewer #2:
Comments:
General comments:
This manuscript does a very nice job of detailing how GSTs might be involved in the detoxification of (S)-(-)-palasonin in Plutella xylostella. The data clearly point to differences between the sensitive and non-sensitive Plutella xylostella, changes in GST activity following exposure to sub lethal amounts of (S)-(-)-palasonin, and changes in the expression of the GSTs.  Further the recombinant GST proteins indicate that GSTd1, GSTd2, GSTs1, and GSTs2 are actively involved in the metabolism.  On the whole, i think the work is well done, it appears time and care was put into the preparation of the manuscript, and finally the figures are simple and easy to interpret.  A few points that will improve the manuscript is the incluse of the methods for the detoxification enzymes (GST, P450, and CarE) in the methods section, correcting the formula used to calculate PLN metabolism, and a consideration of rearranging of figure 3 to group the data by expression profile instead of numerical order.

Response: Thank you for you positive view on our manuscript and your very constructive suggestions. We have re-read the manuscript and made the changes as suggested to improve readability. In the resubmitted version of our manuscript we highlight in red the parts that have been modified. Thanks again for the positive and constructive comments.

Line 12:  “has a good toxic” Good is a very broad qualifier, so maybe saying “competent” better fits your meaning.

Response: We thank the reviewer for this suggestion, we have made this change.

23 & 35: PLN - this term is not previously defined, therefore it needs to be defined or spelled out.

Response: Thank you for reading our manuscript carefully. The manuscript has been revised according to the reviewer's comments.

53:  The term “stomach” should be replaced by midgut.

Response: Thank you for your valuable comments on our paper.We agree that the term stomach is inappropriate for the insect gut, and we have replaced it with ‘ingestion’.

54:   LC50 is in units of mg/L, you should replace that value with molarity, or at minimum include molarity in parenthesis afterward.

Response: Thank you for your time and effort to carefully review our manuscript. However, we prefer to keep the mg/L unit for the median lethal concentration (LC50) data, as this represents the most common format to report LC50 values (e.g., Liu et al., 2010; Cui et al., 2019; Li et al., 2019).

Cui, C.; Yang, Y.; Zhao, T.; Zou, K.; Peng, C.; Cai, H.; Wan, X.; Hou, R. Insecticidal Activity and Insecticidal Mechanism of Total Saponins from Camellia oleifera. Molecules, 2019, 24, 4518.

Li, Y.; Wei, J.; Fang, J.; Lv, W.; Ji, Y.; Aioub, A.A.A.; Zhang, J.; Hu, Z. Insecticidal Activity of Four Lignans Isolated from Phryma leptostachya. Molecules, 2019, 24, 1976.

Liu, M.; Wang, Y.; Wangyang, W. Z.; Liu, F.; Cui, Y. L.; Duan, Y. S.; Wang, M.; Liu, S. Z.; Rui, C. H. Design, synthesis, and insecticidal activities of phthalamides containing a hydrazone substructure. Journal of agricultural and food chemistry, 2010, 58(11), 6858-6863.

101:  References a “leaf dipping method above” but it is not described above.

Response: We apologize for this careless mistake. We have re-checked and thanks to the reviewer for point it out.

112:  You reference the methods for GST, P450 and CarE in Li et al. You need to follow that statement with a brief description (this serves as a method for a reader who may not have access to that article the ability to understand how the data was collected).

Response: We apologize for not clearly presenting the methods. We have re-written this part and aimed to provide a more easily comprehensible description. More detail has been added to the Methods section. The revised version is as shown below:

“For GST, the working solution included 2 μL of enzyme, 2 μL of GSH, 10 μL of 1-chloro-2,4-dinitrobenzene (CDNB, Solarbio, China) and 186 μL sodium phosphate buffer. Then, the absorbance at 340 nm every 1 min for 6 times was measured. For CarE, the working solution was incubated at 25 ℃ for 15 min, which included 5 μL of enzyme, 190 μL of Tris-HCl/CaCl2 (25 mM/1 mM, pH 7.0) and 5 μL of α-NA (0.1 mM). Then, 25 μL fast blue B salt (0.2%) (MP Biochemicals, U.S.A.) and 25 μL sodium dodecyl sulfate (SDS, 0.42%) were added to the mixture and incubated at room temperature for 30 min under dark. The absorbance was recorded at 600 nm. For P450, the working solution was incubated at 30 ℃for 2 h, which included 10 μL of enzyme, 10 μL of nicotinamide adenine dinucleotide phosphate (NADPH, 9.6 mM), 2 mM of p-nitroanisole and 180 μL sodium phosphate buffer (100 mM with 0.5% Triton X-100, pH 7.2). The absorbance was recorded at 405 nm.”

149-151:  Bacterial cells are commonly described as cell “pellet” instead of cell “precipitate”.  I’m thinking this should read, The bacteria were pelleted by centrifugation (8000 g, 4ËšC), then resuspended in PBS.  After sonication the lysate was centrifuged and the target proteins purified from the supernatant by Ni-NTA.  Or something similar.

Response: We appreciate you for the comments and thanks for the good suggestion. We have revised this part.

157:  The abbreviation “CDNB” is used but is not defined.  Please use the molecules full name followed by the abbreviation in parenthesis.

Response: Sorry for the oversight and we have modified. Once again, thank you very much for your comments and suggestions.

159:  not “instilled” were pipetted.

Response: Corrected, thank you!

179:  “dichloromethane mixtures”  should read dichloromethane “aliquots” were added….

Response: Thank you for your comments. We revised the text according to your suggestion.

185:  The formula for the calculation is incorrect.  As written “In this formula, NC represents the GST enzyme control without (S)-(-)-palasonin and CT represents the sample test with inhibitor ((S)-(-)-palasonin.”  If this were correct the peak area without PLN should be near zero, then subtracting the peak area of the sample with PLN would result in a negative value, which is then divided by a near zero number.  This shouldn’t be. I’m assuming that in the experiment NC is without PxGST, and CT is with PxGST.  If this is not the case a much better explanation of the experiment is required, and I cannot accept that experiment as valid.

Response: We seriously apologize for this carelessness and thank you so much for pointing out this issue. We have corrected these errors. Actually, in this formula, NC represents the peak area of control group and CT represents the peak area of (S)-(-)-palasonin treatment group. The difference between the control and treatment groups was that PBS replaced the proteins in the reaction system.

236: “this increment” is awkward,  “this increase” makes more sense.

Response: Thank you. Text was modified based on reviewer’s suggestion.

274:  I appreciate the numerical order to the plots, but I think it would be better if the order were to group the expression patterns.  For example, d3, d4, e5, o3, s1 in a single column or row because they are all up at 6 and 24 hours while down at 12 and 48.

Response: Thank you for your very helpful suggestions to improve our paper. We have made some adjustments to the ordering of the figure panels to keep the plots with the same gene expression trend in the same row (Figure 3.).

284: …molecular weight of c. 25 kD, remove the c.

Response: Thank you. We have deleted the “c.”.

284: The yield of four, but you list 9 proteins.

Response: We apologize for this careless mistake. It has been corrected. Thanks to the reviewer for point it out.

285: c. need to be removed again

Response: It has been deleted. Thank you.

290: Since all the units for the vmax are the same, it might be less distracting to say, the values for vmax (in μmol/min/mg) are….

Response: Thank you for your comment. We agree and revised accordingly.

300:  “as have” should be “as having”

Response: We are sorry for the incorrect expression and have corrected it.

336:  you use icorr, I would write it out.

Response: We are sorry for the misstatement, relevant sentence has been modified. Thank you again for all of your comments.
